# Epigenetic Alterations in Inborn Errors of Immunity

**DOI:** 10.3390/jcm11051261

**Published:** 2022-02-25

**Authors:** Roberta Romano, Francesca Cillo, Cristina Moracas, Laura Pignata, Chiara Nannola, Elisabetta Toriello, Antonio De Rosa, Emilia Cirillo, Emma Coppola, Giuliana Giardino, Nicola Brunetti-Pierri, Andrea Riccio, Claudio Pignata

**Affiliations:** 1Department of Translational Medical Sciences, Università degli Studi di Napoli “Federico II”, 80125 Naples, Italy; roberta.romanomd@gmail.com (R.R.); francescacillo97@gmail.com (F.C.); cristinamoracas@gmail.com (C.M.); nannola.chiara@libero.it (C.N.); betty.toriello@gmail.com (E.T.); antonioderosa06@libero.it (A.D.R.); emiliacirillo83@gmail.com (E.C.); emmacopp24@gmail.com (E.C.); giuliana.giardino@unina.it (G.G.); brunetti@tigem.it (N.B.-P.); 2Department of Environmental Biological and Pharmaceutical Sciences and Technologies, Università degli Studi della Campania “Luigi Vanvitelli”, 81100 Caserta, Italy; laura.pignata@unicampania.it

**Keywords:** epigenetics, DNA methylation, inborn errors of immunity

## Abstract

The epigenome bridges environmental factors and the genome, fine-tuning the process of gene transcription. Physiological programs, including the development, maturation and maintenance of cellular identity and function, are modulated by intricate epigenetic changes that encompass DNA methylation, chromatin remodeling, histone modifications and RNA processing. The collection of genome-wide DNA methylation data has recently shed new light into the potential contribution of epigenetics in pathophysiology, particularly in the field of immune system and host defense. The study of patients carrying mutations in genes encoding for molecules involved in the epigenetic machinery has allowed the identification and better characterization of environment-genome interactions via epigenetics as well as paving the way for the development of new potential therapeutic options. In this review, we summarize current knowledge of the role of epigenetic modifications in the immune system and outline their potential involvement in the pathogenesis of inborn errors of immunity.

## 1. Introduction

Epigenetics is now emerging as an important tool of fine-tuning of gene transcription, and is thus directly implicated in cell maturation and functionality. The collection of genome-wide DNA methylation data has also recently shed new light onto the potential contribution of epigenetics in pathophysiology and, in particular, in the field of immune response. Here, we focus on the association between the pathogenesis of inborn errors of immunity and alterations of epigenetic modifications. In particular, we will summarize well-known disorders or newly identified syndromes in which disturbances of epigenetic machinery may help explain undefined cases and that may, in turn, clarify the contribution of the epigenome to immune system development.

## 2. Physiologic Roles of Epigenetics 

During development, stable and heritable mechanisms, such as histone modifications and DNA methylation, are employed for the functional regulation of gene expression [1,2]. These processes are referred to as “epigenetics”, indicating changes occurring without a direct alteration of the DNA sequence. Under this term are included all the changes exerted via the regulation of chromatin functions and states of activation that are critical for the control of DNA accessibility and transcription. Indeed, the three-dimensional folding of the nuclear genome is tightly linked to the functional DNA-dependent processes of replication and transcription. In particular, DNA replication is a complex and dynamic phenomenon based on the interplay among the epigenetic signature, the transcriptional activity and the structure of chromatin into which DNA is folded and condensed into the nucleus [3]. The double-stranded DNA is wrapped around a core of 8 histone proteins, including two copies of each histone, H2A, H2B, H3, and H4, forming superordinate biomolecular structures, namely nucleosomes, that build up chromatin fibers [4] (Figure 1A). Because of the flexibility of chromatin fibers, target DNA sequences can contact their regulatory elements, even though they are distantly located. Chromosomes segregate into two mutually exclusive types of chromatin, “A” and “B” compartments, including gene-rich active and repressive chromatin, respectively [5,6]. The A compartment is located centrally, whereas the B compartment is typically located peripherally in the nucleus [6]. Besides chromatin structure remodeling, DNA methylation, histone post-translational modifications and non-coding RNAs (ncRNAs) are the key epigenetic factors involved in the dynamics of transcriptional control. 

For DNA methylation, a methyl group is added to the carbon 5 (5meC) of cytosine-followed-by-guanine dinucleotides (CG or CpG sites) by DNA methyltransferases (DNMTs), a family of four enzymes. Unlike in mammals, bacterial methylation may also occur at N4 cytosine (4meC) and N6 adenine (6meA), the latter being the most prevalent in bacteria [7]. 

In mammals, DNMT3A, DNMT3B and DNMT3L catalyze de novo DNA methylation with differential kinetics and patterns during male and female gametogenesis and within cell lineage specification in post-implantation development [8,9], while DNMT1 secures the maintenance of DNA methylation following replication through cell division [10].

In the primordial germ cells and pre-implantation embryo, two waves of extensive erasure involving both passive and active mechanisms occur. Consecutive cell divisions may be followed by passive DNA demethylation, while enzyme members of the Ten-Eleven Translocation (TET) family mediate active demethylation through the oxidation of 5-methylcytosine (5mC) to 5-formylcytosine (5fC), 5-hydroxymethylcytosine (5hmC) and 5-carboxylcytosine (5caC), followed by the replication-dependent dilution of oxidized 5mC or the thymine DNA glycosylase (TDG)-mediated excision of 5fC and 5caC coupled with base excision repair [10].

In general, DNA methylation is high across gene bodies and inter-genic regions, and low at regulatory regions, such as promoters and enhancers. Once established on regulatory regions, methylation can be repressive for transcription because it either directly inhibits the binding of transcription factors or indirectly inhibits the activity of methyl-binding proteins and chromatin modifiers. The methylation of gene bodies is not a repressive mark, but it prevents spurious transcription initiation [11].

Although it is prevalent in mammals, it is worth mentioning that CpG methylation is not the only model of methylation: evidence of non-CpG methylation sites has been found in human embryonic stem cells, induced pluripotent stem cells and brain tissues [12,13]. Hence, novel research to ascertain its role in the maintenance of pluripotency, as well as in the pathophysiology of cancer and neurodegeneration, has developed significantly in recent years [14,15,16]. 

Histone marks, such as histone H3 modifications, correlate with gene expression. For example, histone 3 lysine 4 trimethylation (H3K4me3) and/or histone 3 lysine 27 acetylation (H3K27ac) are active marks, found at active promoters and/or enhancers. They correlate negatively with DNA methylation and positively with gene expression [17]. 

Proper DNA methylation is required for normal human development [18]. Methylation abnormalities may be associated with genetic defects involving *cis*-acting elements or *trans*-acting factors, but can also occur in the absence of obvious genetic changes as primary epimutations; these may represent stochastic or environment-driven errors in the establishment or maintenance of an epigenetic program [19]. Single-locus methylation defects can be a consequence of a variant occurring in *cis*, while, when multiple loci are involved, this may be due to a variant occurring in *trans*. In the latter case, variants in DNMTs or mutations in chromatin modifiers or transcription factors may alter genomic methylation [18].

In cancer, abnormal DNA methylation patterns have been frequently demonstrated, such as the hypermethylation of tumor suppressor gene promoters or the methylation changes of imprinted loci [20].

Recently, through the use of high-throughput screening platforms, an increasing number of disorders have been associated with specific “episignatures”, indicating that DNA methylation analysis may represent a powerful tool for the more accurate classification of diseases with overlapping clinical signs and for categorizing cases with unclear genetic variants [21].

The so-called non-coding RNAs include microRNAs (mRNAs) and long-non-coding RNAs (lncRNAs), both of which are involved in the regulation of gene expression. The former are short molecules that bind to complementary sequences in the 3′ UTR region of the mRNA, directly inhibiting its translation or inducing its degradation. The latter are longer than 200 nucleotides and act by binding to histone modifiers or transcription regulation proteins [22]

Since they play a crucial role in regulating processes such as proliferation, differentiation, development, and apoptosis, it is not surprising that the disruption of their function is also relevant for human diseases, as uncovered by evidence gained in the field of tumorigenesis [23].

## 3. Epigenetics in the Immune System

A growing body of evidence suggests that epigenetic mechanisms, including DNA methylation, play a key role in hematopoiesis, contributing to the differentiation of the hematopoietic stem cell (HSC) into different subsets of immune cells, namely towards the lymphoid and myeloid lineages. Indeed, each cell subset exhibits a unique methylation profile, with remarkable differences between the cells of the myeloid and lymphoid lineages [24].

DNA methylation is increased with lymphoid differentiation but reduced in myeloid differentiation [25] (Figure 1B). Interestingly, in humans, the inactivation of DNA demethylating enzymes TET has been associated with several myeloid malignancies, as myeloproliferation replaces cell differentiation [26]. 

During each step of their development, B cells undergo methylation changes in up to one third of all their genome CpGs. In the early phases of differentiation in the bone marrow, these changes are considered lineage-determining. Non-CpGs demethylation occurs upon B-cell commitment in pre-B2 cells, while CpG methylation changes in effector genes are detected in all other stages of B-cell maturation and activation after B-cell receptor stimulation by antigen binding in the spleen [27].

DNA methylation and histone acetylation are also involved in V(D)J recombination, a process that causes changes in chromatin structure and allows recombination steps through the activity of RAG1/2 enzyme, which recognizes specific signal sequences [28].

As for the T-cell compartment, when the lineage choice of T cells occurs, DNA methylation of the *Cd4* locus is required for its repression in CD8+ cells and its expression in CD4+ cells, as demonstrated in mouse models [29]. In the thymus, DNMT1 interaction with FOXP3 (Forkhead Box P3) transcription factor induces Tregs development. Tregs are a heterogeneous population of CD4-positive T cells characterized by a high expression of CD25 and a low expression of CD127 [30]. After T cell activation, active DNA demethylation is essential for interleukin-2 (IL2) synthesis and for lineage polarization into T helper-1 (Th1), Th2, and Th17 [31,32]. 

DNA methylation plays a critical role in CD4+ T-cell differentiation: DNMT1 loss leads to decreased peripheral T-cell proliferation and the increased expression of cytokines such as IL-2, IL-3, IL-4 and IFNγ, in activated CD4+ (and CD8+) T cells, suggesting a repressive function of DNMT1 towards cytokine production. Under TH2 polarizing conditions, DNMT1 dissociates from the IL4 locus, enabling the demethylation of the locus and the increased expression of IL-4 [33].

The shift to a memory-like phenotype induced in NK cells by some viral infections may also rely on changes in the methylome profiling of promoters of cytokines, including IL13, IL5, and IFN, which become demethylated, as observed in T-cell activation [33,34]. 

The role of DNA methylation machinery has also been described in the mononuclear-phagocyte system during monocyte differentiation into macrophages and their polarization to a “M1” state or an anti-inflammatory “M2” phenotype, as well as in keeping the neutrophil phenotype fully differentiated [35]. 

Extensive mRNA expression profiling has widely demonstrated how hematopoiesis and cell lineage commitment are also accompanied and orchestrated by changes in mRNA signatures [36]. For instance, relevant steps in both T- and B-cell lymphopoiesis rely on gene regulation by specific sets of miRNA [37]. Notably, hematopoiesis also undergoes regulation by lncRNAs that stimulate the proliferation and differentiation of erythroid progenitors by targeting GATA1, TAL1 and KLF1, as well as granulocyte differentiation, thanks to HOTAIRM1, that acts as a regulator of cell cycle [38,39,40]. 

## 4. Epigenetic Alterations in Inborn Errors of Immunity

Since the proper establishment of DNA methylation patterns is necessary for the differentiation of cells of the immune system, the impairment of DNA methylation machinery results in immune dysfunction and diseases. Historically known as primary immunodeficiencies, Mendelian disorders of the immune system are now referred to as Inborn Errors of Immunity (IEI), a more precise and wider definition that takes into account the traditionally known feature of increased susceptibility to infections along with remarkable immune dysregulation and/or hyperinflammation [41,42]. More than 400 genes have been included in the most recent classification of by the International Union of Immunological Sciences [43,44]. In the following sections, we review the potential involvement of epigenetic alterations in the pathogenesis of some inborn errors of immunity, whose features are summarized in Table 1.

### 4.1. Inborn Errors of Humoral Immunity 

Common Variable Immunodeficiency (CVID) is a heterogeneous group of disorders characterized by hypogammaglobulinemia and impaired response to vaccinations. CVID is characterized by marked genetic and phenotypic heterogeneity and monogenic variants have been identified in no more than 10% of patients [45]. Thus, the majority of CVID patients lack a monogenic basis and a polygenic origin may be assumed in most cases. Since a genetic diagnosis of CVID can be achieved only in a small percentage of patients [46], epigenetic alterations, such as DNA methylation and histone modifications, may be theoretically envisioned as potential mechanisms implicated in genetically undefined cases, as a few studies, described below, seem to suggest. 

In the early stages of B cell differentiation, during the transition from pro-B to pre-B cells, an alteration in DNA methylation occurs, especially in intragenic and intronic regions [47] closely associated with transcription factor sites related to B cell development, such as *EBF1*, *E2F*, and *PAX5* [48]. Tallmadge et al. analyzed the transcriptome sequencing of horses affected by CVID, revealing a significant down-regulation in *PAX5* expression. The suspicion of an epigenetic mechanism responsible for this down-regulation was confirmed by the analysis of the epigenomic profile, which revealed a hypermethylation of the *PAX5* enhancer in the bone marrow of CVID-affected horses [49].

However, the most important alterations in DNA methylation are observed in the transition from naïve B cells to germinal center memory and plasma cells. B-cell differentiation is associated with a gradual DNA demethylation [27], with a similar grade of DNA methylation in memory and plasma cells, although these two cell lines have different transcriptional profiles [50]. A study on CVID-discordant monozygotic twins revealed an increase in the DNA methylation of critical B lymphocyte genes, such as *PIK3CD*, *BCL2L1*, *RPS6KB2*, *TCF3* and *KCNN4* in the affected sibling, as compared to the healthy sibling. This hypermethylation, observed in both unswitched- and switched-memory B cells, led to a down-regulation of those genes and, consequently, to B cell dysfunction [51].

In another study, the DNA methylome of CVID patients was compared with that of healthy donors, underpinning the hypothesis that altered demethylation during B cell differentiation may contribute to the pathogenesis of CVID, with a reduction in memory B cells paralleling the degree of demethylation impairment [51].

Immunodeficiency with centromeric instability and facial anomalies syndrome (ICF) is a rare disease caused by biallelic mutations in DNA methyltransferases, characterized by instability of the pericentromeric heterochromatin of chromosomes 1, 9 and 16, peculiar facial anomalies and immune deficiency. The latter may have a variable degree of severity, ranging from complete agammaglobulinemia to decreased levels of single classes of immunoglobulins, lymphopenia, T-cell proliferative response [52,53,54] and, rarely, autoimmunity [52]. Recurrent respiratory and gastrointestinal infections are typical features. 

ICFs are classified according to genetic defects in ICF1, ICF2, ICF3 and ICF4, due to mutations in the *DNMT3B*, *ZBTB24*, *CDCA7* and *HELLS* genes, respectively [55,56,57]. 

As for ICF1, most patients harbor mutations in the catalytic domain of DNMT3B and show hypomethylation of DNA at determined noncoding repetitive sequences and genes located in inactive heterochromatin, causing chromatin decondensation and chromosomal instability [58]. It has been hypothesized that dysregulated DNA methylation underlies an abnormal maturation of B cells and the generation of immunologic memory [59]. Indeed, lymphoblastoid B cell lines from patients with ICF1 show an impaired expression of the genes involved in critical processes such as lymphocyte signaling, maturation and migration. When compared with controls, almost half of these genes appear to be up-regulated. Additionally, the finding of an increased histone trimethylation at lysine-4, H3K4me3 supported this hypothesis [59].

*ZBTB24* is another regulator of hematopoietic development and, being highly expressed in naïve B cells, has a paramount role in B-cell differentiation [60]. In ICF patients with *ZBTB24* mutations, a normal number of total B lymphocytes, as well as naïve and unswitched-memory B cells, has been described, associated with a decrease in switched-memory B cells [53]. 

Finally, the *HELLS* gene, causing ICF4, encodes a lymphoid-specific, ATP-dependent, chromatin-remodeling enzyme, which forms a complex with CDC7A protein, whose gene defect underlies ICF3. Together, they activate chromatin-remodeling activity and, presumably, as in mouse models, exert epigenetic control over B cell development [61]. 

Kabuki syndrome (KS), a rare, multisystemic genetic syndrome associated with an immune disorder, has an estimated prevalence of 1:30,000–1:40,000 individuals. It is characterized by typical facial features, mild-to-moderate developmental delay, cardiac, skeletal and/or renal malformations and immunological abnormalities [62]. Children with KS might share some immune system abnormalities overlapping with CVID, such as hypogammaglobulinemia, increased susceptibility to upper and lower respiratory tract infections and a higher risk of lymphoproliferation [63]. Autoimmune manifestations have also been reported, the most common being autoimmune thrombocytopenia, with or without hemolytic anemia, followed by thyroiditis, celiac disease and vitiligo [64,65]. Seventy percent of KS cases are caused by mutations in the histone methyltransferase *KMT2D* [66], whereas the remaining cases are due to mutations in the histone demethylase *KDM6A* [67]. Both genes contribute to gene expression during embryogenesis. In particular, KMT2D is a lysine H3K4 mono-methyltransferase belonging to the SET domain containing 1/Mixed-Lineage Leukemia (SET1/MLL) protein family, whereas KDM6A acts on H3K27-methylated lysine to remove a repressive mark [68]. Antibody deficiency, as well as a reduction in B cells, total-memory B cells and class-switched-memory B cells, have been detected in KS patients [69].

The immune defects described in KS patients may depend on a loss of H3K4 methylation occurring at crucial transcription factors, dysregulating T and B lymphocyte differentiation. KMT2D loss-of-function might also cause a direct alteration of the antibody maturation, reducing the efficiency of class-switch recombination, while autoimmunity may derive from B-cell tolerance breakage or defective Treg generation [70].

### 4.2. Inborn Errors of Adaptive Immunity

The 22q11.2 Deletion Syndrome (22q11.2 DS) is the most common chromosomal microdeletion disorder. It is characterized by a wide phenotypic spectrum and includes multi-organ defects with congenital heart disease, immunodeficiency, hypoparathyroidism, genitourinary problems, palatal abnormalities, developmental delay and psychiatric symptoms [71]. So far, no single gene has been identified to explain all the features of 22q11.2DS and epigenetic mechanisms have been proposed to explain the clinical variability [72]. The phenotype of 22q11.2 DS could be the sum of the haploinsufficiency of 22q11.2 genes, as well as histone and DNA methylation defects [73]. *TBX1* (T-box 1) is the main candidate gene to explain the disease manifestations and it is involved in chromatin accessibility and transcriptional regulation [74]. *TBX1* was found to co-localize with three H3K4 methyltransferases in ChIP–Western blot analyses of co-immunoprecipitation experiments. In mouse models, *Tbx1* haploinsufficiency is associated with a global reduction in H3K4me1 histone monomethylation levels, causing the differential expression of some protein-coding genes [75]. A genome-wide DNA methylation analysis conducted on 22q11.2DS patients by Rooney et al. [73] led to the identification of 160 differentially methylated CpG probes, retained for the epigenetic signature of the syndrome. Moreover, the DNA methylation profile described was different in patients carrying typical deletions as compared to patients with atypical distal deletions. Identifying the target genes and functional consequences of the histone and DNA methylation alteration in 22q11.2 DS will help to better understand the pathogenesis of the syndrome. 

The dysregulation of miRNAs and lncRNAs due to microdeletion may also partially account for the heterogeneity of the immunological and clinical phenotypes of the syndrome.

Moreover, a reduced function of miR1857, among others, may contribute to a decreased expression of Bruton’s tyrosine kinase (Btk) and marginal-zone B1 protein (Mzb1), thus explaining a subsequent reduction in memory B cells [72].

Schimke immuno-osseous dysplasia (OMIM 242900) is an autosomal recessive disorder, due to mutations in *SMARCAL1* gene encoding SWI/SNF-related, matrix-associated, actin-dependent regulator of chromatin, subfamily A like 1, a chromatin-remodeling enzyme. The function of SMARCAL1 is to regulate transcription through chromatin remodeling [76]. The clinical phenotype includes: dysmorphic features, short stature with skeletal abnormalities, such as spondyloepiphyseal dysplasia and exaggerated lumbar lordosis; and arteriopathy. Impaired kidney function and immune deficiency consisting in recurrent bacterial, viral, or fungal infections have also been reported [77]. Laboratory tests show lymphopenia, absent mitogen-induced proliferation response, reduced CD8 and CD3/CD4 T cells [78]. Loss-of-function mutations in *SMARCAL1* may lead to genome instability, since the enzyme recognizes transitions from single- to double-stranded DNA. 

### 4.3. Inborn Errors of Innate Immunity

No studies of the methylation changes that could occur in this subgroup of disorders have been conducted. However, given that the signaling pathways affected in these diseases, such as that of TLR4, have been described in some cases to cause modifications in DNA methylation, it is conceivable to hypothesize that DNA methylation may potentially exert a mechanistic role in the pathogenesis of undefined disorders or, putatively, in modulating the natural history.

Mendelian susceptibility to Mycobacterial disease (MSMD) is a rare inherited condition characterized by selective predisposition to clinical disease caused by weakly virulent mycobacteria, such as bacillus Calmette–Guerin vaccines and non-tuberculous environmental mycobacteria [79] in otherwise healthy patients with no overt abnormalities in routine hematological and immunological functionality. MSMD patients are also at higher risk of tuberculosis, salmonellosis, candidiasis and, more rarely, to infections with other intra-macrophagic bacteria, fungi, or parasites [80]. Nine MSMD-causing genes, including seven autosomal (*IFNGR1*, *IFNGR2*, *STAT1*, *IL12B*, *IL12RB1*, *ISG15* and *IRF8*) and two X-linked (*NEMO* and *CYBB*) genes, have been described so far; all are involved in IFN-γ-dependent immunity [81]. Pacis et al. showed that the *Mycobacterium tuberculosis* infection of dendritic cells induces rapid loss of DNA methylation at distal enhancers that activate master immune transcription factors (including nuclear factor-kB and members of the Interferon Regulatory Factor family), suggesting an important role for DNA methylation in regulating innate immune responses [82,83]. 

### 4.4. Inborn Errors of Immunity with Immune Dysregulation

Bi-allelic loss-of-function variants in *TET2* in humans have been associated with immunodeficiency and autoimmune lymphoproliferative syndrome (ALPS)-like phenotypes with remarkable predisposition to lymphoma [84]. TET2 is a crucial epigenetic regulatory factor in hematopoietic cells, facilitating demethylation by oxidizing 5-methylcytosine (5mC) to 5-hydroxymethylcytosine (5hmC) and other oxidation products. Loss-of-function mutations in TET2 are responsible for DNA methylation increases in hematologic cells, thus accounting for the failure of the controlled development of B cells and the expansion of double-negative T cells [85]. In Tregs, TET is implicated in the stability of Foxp3 molecules. The haploinsufficiency of TET2 is related to hematological neoplasia. However, it should be mentioned that TET2 mutations also occur in healthy subjects with clonal hematopoiesis, implying that they are sufficient to induce cancer alone [86,87]. Extrinsic factors, namely infections through hyperinflammation, seem to be co-factors in carcinogenesis.

The activity of activation-induced cytidine deaminase (AID) is hampered in TET2-/- mice, leading to abnormal demethylation. Altogether, these changes impair the transcription of genes critical for germinal center exit, antigen presentation and the differentiation of germinal center B cells, concurring with the development of diffuse large B-cell lymphomas. Therefore, it is conceivable to presume that TET2 has a crucial role in cell proliferation and differentiation [87].

TET proteins are also essential for specific points of B cell development, such as the transition from pro-B to pre-B and the differentiation of plasma cells [88]. 

## 5. Conclusions

Gene expression in the immune system is tightly regulated by epigenetic processes, including DNA methylation, chromatin remodeling and histone modifications, that orchestrate development, maturation and cell lineage commitment. In line with this, specific DNA methylation signatures and histone modification patterns can be detected for each cell population. 

Next-generation sequencing technologies have enabled the identification of several new forms of IEI, surprisingly changing the scenario and expanding the knowledge of their molecular basis. Nonetheless, a genetic etiology still needs to be elucidated for many of them; hence, it is reasonable that alterations to the epigenetic mechanisms that control the transcription of genes involved in immune response may contribute to the pathogenesis of at least some of these disorders.

In addition, although most genetic IEIs are paradigmatic examples of monogenic disorders, a broad spectrum of severity and clinical phenotypes is widely recognized. Therefore, epigenetic signatures may be implicated in the regulation of disease expressivity and penetrance, possibly expanding the phenotype.

## Figures and Tables

**Figure 1 jcm-11-01261-f001:**
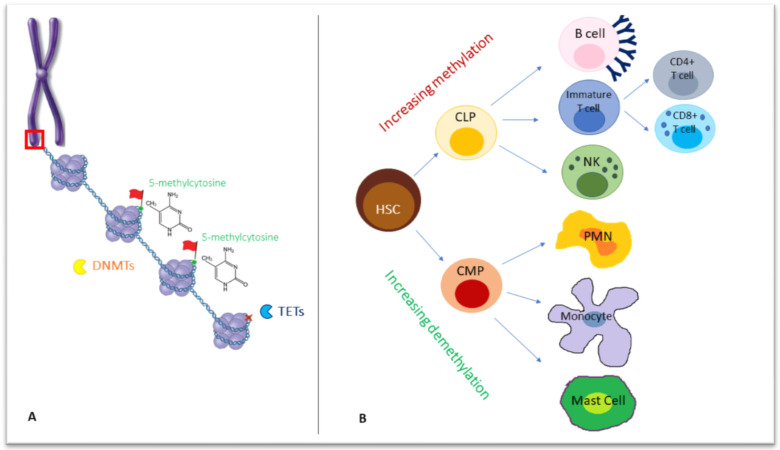
(**A**) Chromosomes formed by chromatin fibers organized into nucleosomes, in which they are wrapped around eight histone proteins (as shown in the red box, zooming in chromosome structure). On a deeper level, DNA methyltransferases (DNMTs) act on double-stranded DNA, adding a methyl group to the carbon 5 (5meC) of cytosine-followed-by-guanine dinucleotides, while Ten-Eleven Translocation (TET) enzymes are responsible for demethylation, removing a 5meC (red X in the figure). (**B**) The differentiation from a common progenitor, a hematopoietic stem cell, to lymphoid and myeloid lineage, is accompanied by a wave of increasing methylation or demethylation, respectively. DNMTs: DNA methyltransferases, TET: Ten-Eleven Translocation; HSC: hematopoietic stem cell, CLP: common lymphoid progenitor, CMP: common myeloid progenitor, PMN: polymorphonucleate.

**Table 1 jcm-11-01261-t001:** Representative gene defects causing epigenetic changes and immunological alterations within defined syndromes.

Humoral Immunity	Disorder	Altered Epigenetic Mechanism	Genes	Major Immunological Alteration	
	CVID	DNA methylation	*PAX5*, *PIK3CD*, *BCL2L1*, *RPS6KB2*, *TCF3*, *KCNN4*	Agammaglobulinemia, impaired response to vaccines, autoimmunity, CLD, enteropathy	
	ICF1	DNA methylation	*DNMT3B*	Agammaglobulinemia or hypogammaglobulinemia, recurrent infections	
	ICF2	DNA methylation	*ZBTB24*	Agammaglobulinemia or hypogammaglobulinemia, recurrent infections	
	ICF3	DNA methylation	*CDCA7*	Agammaglobulinemia or hypogammaglobulinemia, recurrent infections	
	ICF4	DNA methylation	*HELLS*	Agammaglobulinemia or hypogammaglobulinemia, recurrent infections	
	KS1	Histone modification	*KMT2D*	Hypogammaglobulinemia, autoimmune cytopenia	
	KS2	Histone modification	*KDM6A*	Hypogammaglobulinemia, autoimmune cytopenia	
**Adaptive immunity**					
	22q11.2 DS	DNA methylation Non-coding RNAs	*TBX1*	Lymphopenia, recurrent infections, autoimmunity	
	Schimke immuno-osseous dysplasia	Chromatin remodeling	*SMARCAL1*	Lymphopenia, recurrent infections	
**Immune dysregulation**					
	TET2 loss-of-function	DNA methylation	*TET2*	Hepatosplenomegaly, lymphadenopathy, autoimmunity	

## Data Availability

Not applicable.

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
