# Peer review of "Epigenetic Alterations in Inborn Errors of Immunity"

_jcm, 2022, doi:10.3390/jcm11051261_

Round 1

Reviewer 1 Report

General comments.

1) The article only deals with a few epigenetic modifications, such as chromatin remodeling, histone modification and CpG DNA methylation (the explanation of this terms in the introduction and their role in the IEIs as the main topic).

However, there are other epigenetic factors (such as various other DNA modifications, modifications of various RNAs, superordinate biomolecular structures (such as nucleosomes, TADs) etc.). I think that this fact should be mentioned in the introduction, as further only the terms chromatin remodeling, histone modification and CpG DNA methylation are being dealt with exclusively.

2) FYI: There is already a review, that in addition, to the three epigenetic factors discussed in this review, also deals with the role of RNAs in IEIs. (Camacho-Ordonez N, Ballestar E, Timmers HTM, Grimbacher B. What can clinical immunology learn from inborn errors of epigenetic regulators? J Allergy Clin Immunol. 2021 May;147(5):1602-1618. doi: 10.1016/j.jaci.2021.01.035. Epub 2021 Feb 17. PMID: 33609625.)

3) For the sake of precedence, any already known findings about non-GpG methylations, that could be relevant for the IEIs, should be mentioned in the introduction.

For instance, in the article that is also cited in the introduction to this review, the observation of changing non-CpG methylation patterns in human b cell differentiation is reported, which could therefore possibly also be relevant for the immune system. (References 18. Kulis M, Merkel A, Heath S, Queirós AC, Schuyler RP, Castellano G, Beekman R, Raineri E, Esteve A, Clot G, Verdaguer-Dot N, Duran-Ferrer M, Russiñol N, Vilarrasa-Blasi R, Ecker S, Pancaldi V, Rico D, Agueda L, Blanc J, Richardson D, Clarke L, Datta A, Pascual M, Agirre X, Prosper F, Alignani D, Paiva B, Caron G, Fest T, Muench MO, Fomin ME, Lee ST, Wiemels JL, Valencia A, Gut M, Flicek P, Stunnenberg HG, Siebert R, Küppers R, Gut IG, Campo E, Martín-Subero JI. Whole-genome fingerprint of the DNA methylome during human B cell differentiation. Nat Genet. 2015 Jul;47(7):746-56. doi: 10.1038/ng.3291. Epub 2015 Jun 8. PMID: 26053498; PMCID: PMC5444519.).

Specific comments.

! I would not rephrase the sentences in such a way, when they have longer in-text matches with other articles! Direct citation might be necessary for this.

Quote from:

References 1. Ikegami, K., et al. "In order to achieve the proper temporal and spatial regulation of these genes throughout development, a set of epigenetic mechanisms are employed, which includes histone modifications and DNA methylation."

Your text:

Lines 34-36. During development, stable and heritable mechanisms, such as histone modifications and DNA methylation, are employed in order to achieve the proper temporal and spatial regulation of gene expression.

Line 54. Here at the beginning of the paragraph, possibly specify that there are several DNA methylation variants (not only of cytosine, where there is also more than one different methylation possibility (4mC)). Further in the text it is only about 5mC methylation.

Line 56. Also specify here that the methyltransferases in mammals are meant.

Line 90. unnecessary _ underscore character?

Line 239. unnecessary “ “ space character?

Author Response

Point 1: The article only deals with a few epigenetic modifications, such as chromatin remodeling, histone modification and CpG DNA methylation (the explanation of this terms in the introduction and their role in the IEIs as the main topic).

However, there are other epigenetic factors (such as various other DNA modifications, modifications of various RNAs, superordinate biomolecular structures (such as nucleosomes, TADs) etc.). I think that this fact should be mentioned in the introduction, as further only the terms chromatin remodeling, histone modification and CpG DNA methylation are being dealt with exclusively.

Response 1: We thank the Reviewer for the comment and mentioned other epigenetic factors in both paragraph 2 and in paragraph 3.

Line 51-52: “nucleosomal particles” has been replaced by “superordinate biomolecular structures, namely nucleosomes.”

Lines 114-123: “the so-called non-coding RNAs include microRNAs (miRNAs) and long-non-coding RNAs (lncRNAs), both involved in regulation of gene expression. The former are short molecules that bind to complementary sequences in the 3′ UTR region of the mRNA, directly inhibiting its translation or inducing its degradation. The latter are longer than 200 nucleotides and act by binding to histone modifiers or transcription regulation proteins [22]. Since they play a crucial role in regulating processes like proliferation, differentiation, development and apoptosis, it is not surprising that also the disruption of their function is relevant for human diseases, as uncovered by evidence gained in the field of tumorigenesis [23]”.

Lines 168-174: “Extensive miRNA expression profiling has widely demonstrated how hematopoiesis and cell lineage commitment are accompanied and orchestrated by changes in miRNA signatures [36]. For instance, relevant steps in both T and B cell lymphopoiesis rely on gene regulation by specific sets of miRNA [37]. Of note, hematopoiesis undergoes regulation also by lncRNAs that, for instance, stimulate proliferation and differentiation of erythroid progenitors by targeting GATA1, TAL1, and KLF1 and granulocyte differentiation too, thanks to HOTAIRM1, regulator of cell cycle [38-40]”.

Point 2: FYI: There is already a review, that in addition, to the three epigenetic factors discussed in this review, also deals with the role of RNAs in IEIs. (Camacho-Ordonez N, Ballestar E, Timmers HTM, Grimbacher B. What can clinical immunology learn from inborn errors of epigenetic regulators? J Allergy Clin Immunol. 2021 May;147(5):1602-1618. doi: 10.1016/j.jaci.2021.01.035. Epub 2021 Feb 17. PMID: 33609625.)

Response 2: We thank the Reviewer for the suggestion and added a further mention to the role of RNA in IEI in paragraph 4, lines 302-306.

“Dysregulation of miRNAs and lncRNAs due to microdeletion may partially account for the heterogeneity of immunological and clinical phenotype of the syndrome.

Moreover, a reduced function of miR1857, among others, may contribute to a decreased expression of Bruton’s tyrosine kinase (Btk) and marginal zone B1 protein (Mzb1), thus explaining a subsequent reduction in memory B cells [72]”.

Point 3: For the sake of precedence, any already known findings about non-GpG methylations, that could be relevant for the IEIs, should be mentioned in the introduction. For instance, in the article that is also cited in the introduction to this review, the observation of changing non-CpG methylation patterns in human b cell differentiation is reported, which could therefore possibly also be relevant for the immune system. (References 18. Kulis M, Merkel A, Heath S, Queirós AC, Schuyler RP, Castellano G, Beekman R, Raineri E, Esteve A, Clot G, Verdaguer-Dot N, Duran-Ferrer M, Russiñol N, Vilarrasa-Blasi R, Ecker S, Pancaldi V, Rico D, Agueda L, Blanc J, Richardson D, Clarke L, Datta A, Pascual M, Agirre X, Prosper F, Alignani D, Paiva B, Caron G, Fest T, Muench MO, Fomin ME, Lee ST, Wiemels JL, Valencia A, Gut M, Flicek P, Stunnenberg HG, Siebert R, Küppers R, Gut IG, Campo E, Martín-Subero JI. Whole-genome fingerprint of the DNA methylome during human B cell differentiation. Nat Genet. 2015 Jul;47(7):746-56. doi: 10.1038/ng.3291. Epub 2015 Jun 8. PMID: 26053498; PMCID: PMC5444519.)

Response 3: We agree with the Reviewer and mentioned the process in both paragraph 1 and paragraph 2, as follows:  

Lines 84-89 “although prevalent in mammals, it is worth to mention that CpG methylation is not the only model of methylation: evidence of non-CpG methylation sites has been found in human embryonic stem cells, induced pluripotent stem cells and brain tissues [12,13]. Hence, novel research to ascertain its role in the maintainance of pluripotency as well as in pathophysiology of cancer and neurodegeneration is developing fervently in recent years [14-16]”.

Lines 137-140 “Non-CpGs demethylation occurs upon B-cell commitment in pre-B2 cells while CpGs methylation changes in effector genes are detected in all other stages of B-cell maturation and activation after B-cell receptor stimulation by antigen binding in the spleen [27]”.

Point 4, Line 34-36: ! I would not rephrase the sentences in such a way, when they have longer in-text matches with other articles! Direct citation might be necessary for this.

Quote from: References 1. Ikegami, K., et al. "In order to achieve the proper temporal and spatial regulation of these genes throughout development, a set of epigenetic mechanisms are employed, which includes histone modifications and DNA methylation.

Your text:Lines 34-36. During development, stable and heritable mechanisms, such as histone modifications and DNA methylation, are employed in order to achieve the proper temporal and spatial regulation of gene expression.

Response 4: the sentence has been rephrased as follows: “During development, stable and heritable mechanisms, such as histone modifications and DNA methylation, are employed for functional regulation of gene expression [1]” and the proper quotation has been added [Shiota et al. 2004].

Point 5: Line 54. Here at the beginning of the paragraph, possibly specify that there are several DNA methylation variants (not only of cytosine, where there is also more than one different methylation possibility (4mC)). Further in the text it is only about 5mC methylation. Line 56. Also specify here that the methyltransferases in mammals are meant.

Response 5: The paragraph has been edited as follows:

Line 56, now line 66: the words “in mammals” have been added.

Lines 58-59, now line 63-65: “unlike mammals, bacterial methylation may also occur at N4 cytosine (4meC) and N6 adenine (6meA), the latter being the most prevalent in bacteria [7]”.

Point 6: Line 90. unnecessary _ underscore character? Line 239. unnecessary “ “ space character?

Response 6: typos have been corrected.

Reviewer 2 Report

The authors describe in a clear and concise manner the role of epigenetics in physiologic state in the immune system and how its alterations are linked to inborn errors of immunity. The flow of the review is nicely developed, as it comments on the different immune cell types in physiology and the same scheme is applied later when describing different diseased conditions. 

As a minor comment, the quality of the Figure 1 should be improved as it comes out pixelized.

Author Response

Point 1: the quality of the Figure 1 should be improved as it comes out pixelized.

Response 1: the quality of the figure has been improved. Now it should be ok.